# Perfectionism in Children and Adolescents with Eating-Related Symptoms: A Systematic Review and a Meta-Analysis of Effect Estimates

Audrey Livet [1,2,3,*], Xavier Navarri [1,2], Philippe Pétrin Pomerleau [1], Sébastien Champagne [4], Fakir Md Yunus [5], Nicholas Chadi [1,6,7], Gail McVey [8,9] and Patricia Conrod [1,2]

1   CHU Sainte-Justine Research Centre, Montreal, QC H3T 1C5, Canada
2   Department of Psychiatry and Addiction, Faculty of Medicine, Université de Montréal, Montréal, QC H3T 1J4, Canada
3   Centre Hospitalier Universitaire de Montréal, Montréal, QC H2X 3E4, Canada
4   Bibliothèque du CHUM, Centre Hospitalier de l'Université de Montréal, Montréal, QC H2X 1N6, Canada
5   Department of Psychology and Neuroscience, Faculty of Science, Dalhousie University, Halifax, NS B3H 4R2, Canada
6   Sainte-Justine Hospital, Montréal, QC H3T 1C5, Canada
7   Department of Pediatrics, Faculty of Medicine, Université de Montréal, Montréal, QC H3T 1J4, Canada
8   Eating Disorders Ontario, University Health Network, Toronto, ON M5G 2C4, Canada
9   Dalla Lana School of Public Health, Social & Behavioural Health Sciences Division, University of Toronto, Toronto, ON M5T 3M7, Canada
*   Correspondence: audrey.livet.chum@ssss.gouv.qc.ca

**Abstract:** Background: Over 40 years of research implicates perfectionism in eating disorders in childhood and adolescence. However, the nature of this relationship remains understudied. To address this gap, we performed a systematic review and a meta-analysis to quantify the magnitude of the associations between perfectionism (i.e., unidimensional perfectionism, perfectionistic strivings, and perfectionistic concerns) and eating-related symptoms during childhood and adolescence. Methods: The literature search was conducted using five electronic databases in accordance with PRISMA guidelines: MEDLINE, Embase, CINAHL Complete, APA PsycINFO, and EMB Reviews. A total of 904 studies were identified; a total of 126 were included in the systematic review, and 65 in the meta-analysis (N = 29,268). Sensitivity analyses were also carried out to detect potential differences in age and clinical status. Results: All the associations we investigated were both significant and positive. Small effect sizes were found between eating global scores and unidimensional perfectionism, perfectionistic strivings, and perfectionistic concerns ($r_{es}$ = 0.19, $r_{es}$ = 0.21, $r_{es}$ = 0.12, respectively) and remained significant in each age group in both clinical and community samples. Perfectionistic concerns were moderately associated with all eating measures, especially in community samples and samples with a mean age under 14. Conclusions: Psychological interventions specially designed to target perfectionistic concerns in the early stages of development may help prevent the onset or reduce the intensity of eating-related symptoms during childhood and adolescence.

**Keywords:** perfectionism; anorexia nervosa; bulimia nervosa; eating disorders; adolescence

## 1. Introduction

Mental illness in general is on the rise among adolescents [1]. Adolescence represents a critical period in the development of eating disorders (ED) [2–9]. In clinical practice and research, eating-related symptoms are often addressed using psychometric scales, self-report questionnaires, or clinical interviews. Most tools use some Likert scale-type items and explore eating-related thought patterns or attitudes in a dimensional way. Adolescence is a critical developmental period involving new challenges and

distressing experiences, where individuals are striving towards high personal standards and concerns surrounding the body [10,11]. These may be in part influenced by media and peer pressure inciting youths to conform to certain physical ideals in order to gain acceptance [10]. Such concerns can be drivers of negative eating concerns (EAC), body dissatisfaction (BD), weight and shape concerns (WSC), fear of maturity, ineffectiveness, drive for thinness (DT), interoceptive awareness (IA), and even internalization of thin ideal (ITI) [12–14]. Additionally, symptoms, such as binge eating (BE), dietary restraint (RT), loss of control, compulsive exercising, or orthorexia are also prevalent during adolescence [8,14,15].

Perfectionism consists of a combination of exceedingly high and unrealistic personal standards and overly self-critical evaluations. According to a recent meta-analysis, college students, in particular, point to an increasing prevalence of perfectionism over the last 30 years [16]. There are two current conceptualizations of perfectionism. One views perfectionism as a unidimensional construct, whereas the other one views it as a multidimensional construct, including a range of subdimensions that can be collectively captured in a two-factor model comprising perfectionistic strivings (PS) and concerns (PC) [17,18]. Perfectionistic strivings designate the pursuit of unrealistic objectives, whereas perfectionistic concerns refer to the fear of making mistakes and being judged negatively by others. A recent meta-analysis of adult samples revealed significant associations of perfectionistic strivings and perfectionistic concerns with various forms of psychopathology [19]. This meta-analytic evidence revealed the equal magnitude of the associations of PC and PS with Eds symptoms [19]. Strong and positive effects sizes were observed between anorexia and some dimensions of perfectionism: concern over mistakes (k = 3, r = 0.92), socially prescribed perfectionism (k = 1, r = 0.78), personal standards (k = 3, r = 0.44), self-oriented perfectionism (k = 1, r = 0.83), and organization (k = 3, r = 0.41) [19]. To date, no such meta-analysis has been conducted in children and adolescent samples. Among adolescents, perfectionism appears to be associated with higher levels of eating disorder symptoms in both clinical [20,21] and community samples [22,23]. Not surprisingly, perfectionism has been found to be associated with the onset and maintenance of Eds [19,23]. Perfectionism is also reported to have a prospective influence on the development of disordered eating based on longitudinal research of community and clinical adolescent samples [24–27]. To date, there exist some theoretical models that point towards perfectionism as being a risk factor for eating disorder symptoms during adolescence [22,23,26]. Perfectionism may serve as a risk factor for eating disorders if the perfectionistic tendencies are exhibited or expressed in the form of doubts and concerns and/or heightened attunement to rigid rules and expectations surrounding food, weight, and shape [28].

A recent narrative review, including 79 studies, has investigated the association of perfectionistic strivings and concerns with eating symptoms during adolescence [29]. The number of associations reported from this review is consistent with the body of adult literature, which suggests a greater strength of the relationship between PC and eating disorders symptoms [19]. However, this narrative review based its conclusions on the number of associations found in the literature without consideration for their magnitude. Considering the differences in the strength of the associations between the two subdimensions across a range of psychopathologies, including EDs in adults, it is warranted to study specific expressions of eating disorders using a meta-analytic approach. Additionally, this narrative review only reports associations from adolescent samples and does not investigate any data from childhood. While associations appeared to be stronger in clinical and older samples [21,30], there is no evidence of such a trend in this review. This narrative review makes it impossible to ascertain the strength of these associations during early development and if they differ in accordance with age or clinical status.

First, the purpose of the present study was to build on the existing body of literature with a rigorous systematic review and meta-analysis to gauge and corroborate available evidence underlying the association between perfectionism and eating-related symptoms during childhood and adolescence. Second, attention was given to investigating if these associations were clinically sensitive by stratifying extracted samples. Third, to reach our aims, the current work also assesses the strength of these associations across samples with a mean age under 14 years old (childhood to early adolescence) versus a mean age above 14 years old (middle to late adolescents).

We hypothesized that (a) unidimensional perfectionism, perfectionistic concerns, and perfectionistic strivings would be positively associated with all investigated eating-related symptoms; (b) there will be differences in the magnitude of the associations of both clinical and community samples; and (c) associations are expected to be different between studies depending on whether the samples include children and early adolescents (i.e., above 14 years of age) or middle-to-late adolescents (i.e., over 14 years of age).

## 2. Materials and Methods

### 2.1. Literature Search Strategy

This review adhered to the 2020 Preferred Reporting Items for Systematic Reviews and Meta-Analyses (PRISMA) guidelines [31] (Table S1, supplementary). The protocol of this systematic review is registered on PROSPERO (CRD: CRD42023393732). The following databases were systematically searched to identify the most pertinent studies: MEDLINE (Ovid), Embase (Ovid), CINAHL Complete (EBSCOHost), APA PsycINFO (Ovid), and EBM Reviews (Ovid). The latter encompasses the Cochrane Database of Systematic Reviews, ACP Journal Club, Database of Abstracts of Reviews of Effects, Cochrane Clinical Answers, Cochrane Central Register of Controlled Trials, Cochrane Methodology Register, Health Technology Assessment, and NHS Economic Evaluation Database.

Search strategies were designed by the first author and a librarian to ensure the validity and completeness of the literature search. The search strategies were also peer-reviewed by a senior librarian. The initial search strategy was duplicated in the other databases with little to no adaptation. A detailed description of the search strategies is included in the supplemental material (Table S2, supplementary). We imposed no language, study types, or other restrictions on any of our searches. The initial comprehensive literature search was run during Summer 2019, and an update was made during Summer 2020. Due to the increased number of studies on the current topic during the last two years, the last update was performed on 9 December 2022 for all sources. As a complement to this search, we performed backward searches by manually examining references of eligible studies for any original articles that were not identified by the computerized literature search. Duplicates were removed in EndNote by the librarian (SC) using the method reported by Bramer [32].

### 2.2. Inclusion and Exclusion Criteria

The selection of the articles was discussed by consensus by three authors (AL, XN, and PPP) using the Covidence software [33]. Covidence is a web-based collaboration software platform that streamlines the production of systematic and other literature reviews. Studies were eligible for inclusion in our systematic review if they met the following three inclusion criteria.

First, we only included studies whose participants were children and adolescents with a mean age under 18 years old. Both community and clinical samples of adolescents were included.

Second, studies were only included if they assessed eating-related symptoms using validated scales or structured interviews. Thus, this included clinical samples that could report diagnoses based on previous versions of the DSM. All studies assessed perfectionism using validated scales. When perfectionism was assessed using behavioral or experimental tasks, studies were excluded.

Third, we also excluded samples with elite athletes, such as ballet dancers, due to some findings indicating that perfectionism and eating concerns may interact differently in this population, making it necessary to consider additional parameters, such as the type of sport and the skill level [34–37]. Although elite athletes may still be included in the community samples, their relative proportion should be insignificantly small in those age groups and methods of recruitment and, therefore, should have little to no effects on the analyses considering the size of our total sample.

We used the Newcastle–Ottawa Scale to assess the quality of longitudinal studies [38]. Furthermore, we adopted the modified version of the Newcastle–Ottawa Scale for cross-sectional studies to assess the quality of included cross-sectional studies [39]. The only restriction that was placed on study types was that we did not include any case report study or single case study, including only one participant. No restrictions were placed on study characteristics regarding gender or ethnicity. Studies from any nation and any time period were considered relevant. All the included articles were in English or French due to the bilingual ability of our research team.

The inclusion criteria for our meta-analysis were similar, with only two additional criteria to address statistical concerns. Studies were categorized as eligible if they contained any statistical information sufficient for extraction (i.e., a measure of association between perfectionism and any eating-related symptoms). Data extraction was conducted by AL and PPP. We also contacted authors to clarify whether there was an overlap in the respective samples to validate the independence of certain included studies. We contacted the primary and the senior authors for any studies where the reported information was insufficient to compute the effect sizes.

### 2.3. Statistical Analysis

Pearson correlation coefficients ($r_{es}$) were used as the effect sizes for the analyses. Using random-effects models, we first performed analyses between the three selected types of perfectionism (i.e., UNI, PS, and PC) and the Eating global score using a validated measurement scale (e.g., the total score on the Eating Disorder Inventory). Then, we performed analyses between all three types of perfectionism and eating-related symptoms (subscales scores). False discovery rate (FDR) corrected *p*-values were calculated for eating-related symptoms and the Eating global score to control for multiple testing for each type of perfectionism. We calculated missing effect sizes using the equations proposed by Lipsey and Wilson [40]. We performed inverse-variance weighted random-effect models for aggregated studies based on eating measures and types of perfectionism using the Metafor package in R [41]. The between-study variability was estimated using the Sidik–Jonkman method because it calculates heterogeneity estimates more accurately than the Dersimonian–Laird method [42]. Statistical heterogeneity was calculated using the $I^2$ statistic.

Additionally, we performed an age-stratified sensitivity analysis between individuals under and over 14 years old. First, the determination of this age is based on the mean age calculated from all included studies which were equal to 14 years old. Second, some studies in the literature indicate that the age of 14 delimits the transition from early adolescence to middle adolescence [43,44]. Third, the lifetime prevalence rates of EDs have been demonstrated to slightly increase from age 14 to 17 [3]. The purpose of grouping studies in terms of their participants' mean age (i.e., below or above 14 years of age) was to observe differences in terms of associations between age groups. All samples with no indication of any mean age were retrieved from any analysis. Finally, we evaluated potential differences in the perfectionism-eating-related symptoms relationships between community and clinical samples. All samples with a mixed population (i.e., including both clinical samples and community samples without specifying eating of perfectionism scores for each population) were retrieved from any analysis.

### 2.4. Publication Bias

Due to studies with effect sizes that are inconsistent with the direction of the hypothesis and are less likely to be published, which can ultimately inflate the meta-analytic summary effect sizes, we evaluated the risk of potential publication bias (i.e., dependence between effect sizes and samples in a meta-analysis) in our associations [45,46]. Publication bias was assessed by three methods for all performed meta-analyses, for which we had more than ten studies by (a) a visual inspection of the funnel plots symmetry, (b) both Egger's linear regression and Begg's rank correlation tests based on the number of original studies available, and (c) the trim-and-fill method [46,47]. Visual inspection and association tests are intended to assess the asymmetry of the funnel plot, which is an indicator of dependence between effect sizes and sample sizes. Such dependence may occur when studies with effect sizes that do not support the hypothesis are not published. Instead, the trim-and-fill method imputes the presence of studies with asymmetry in the funnel plot.

## 3. Results

### 3.1. Systematic Search and Meta-Analysis Results

A total of 1709 articles were retrieved through the search of the computerized databases and examination of cross-references by three authors. A total of 487 articles were retrieved from MEDLINE, 496 from Embase, 44 from EBM Reviews, 401 from PsycInfo, and 260 from CINAHL Complete. We also identified 21 articles by manually examining the references of eligible studies. In total, 126 articles were included as a result of the systematic review procedure (Figure 1) following the removal of duplicates, articles other than original research articles (e.g., meta-analysis, review, letter to the editor, abstract of conference, etc.), articles on different populations, articles not using validated scales, papers irrelevant to the present review and articles published in another language than English or French. Participant demographics and study characteristics are provided in Table S3 in the supplementary materials available online.

Unidimensional measures were used in 86 included studies, whereas the multidimensional approach was employed in 87 included studies. For a list of measurement scales used to assess perfectionism during childhood and adolescence, see the supplementary materials Table S5. Most of the unidimensional studies used the Eating Disorder Inventory-Perfectionism Scale [48] (EDI-P, in 25 studies), whereas studies on PC or PS mostly used the Frost Multidimensional Perfectionism Scale [17] (FMPS, in 25 studies), the Children and Adolescent Perfectionism Scale [49] (CAPS, in 24 studies), or the Hewitt and Flett Multidimensional Perfectionism Scale [50] (HFMPS, in four studies).

In total, 65 articles (N = 29,268) were included in the quantitative study (Tables S3 and S4, supplementary). A total of 13 articles included clinical participants (N = 2169, mean age = 14.76 (1.42), girls = 87%), and 55 included community children and adolescents (N = 27,099, mean age = 14.07 (1.83), girls = 69%). The extracted data allowed the analysis of the relationships between unidimensional perfectionism and eight measure of eating-related symptoms (i.e., eating global score, dietary restraint, interoceptive awareness, binge eating, drive for thinness, weight and shape concerns, internalization of the thin ideal, and body dissatisfaction), between both perfectionistic concerns, perfectionistic strivings, and seven measure of eating-related symptoms (i.e., eating global score, binge eating, dietary restraint, eating concerns, weight and shape concerns, body dissatisfaction, and drive for thinness).

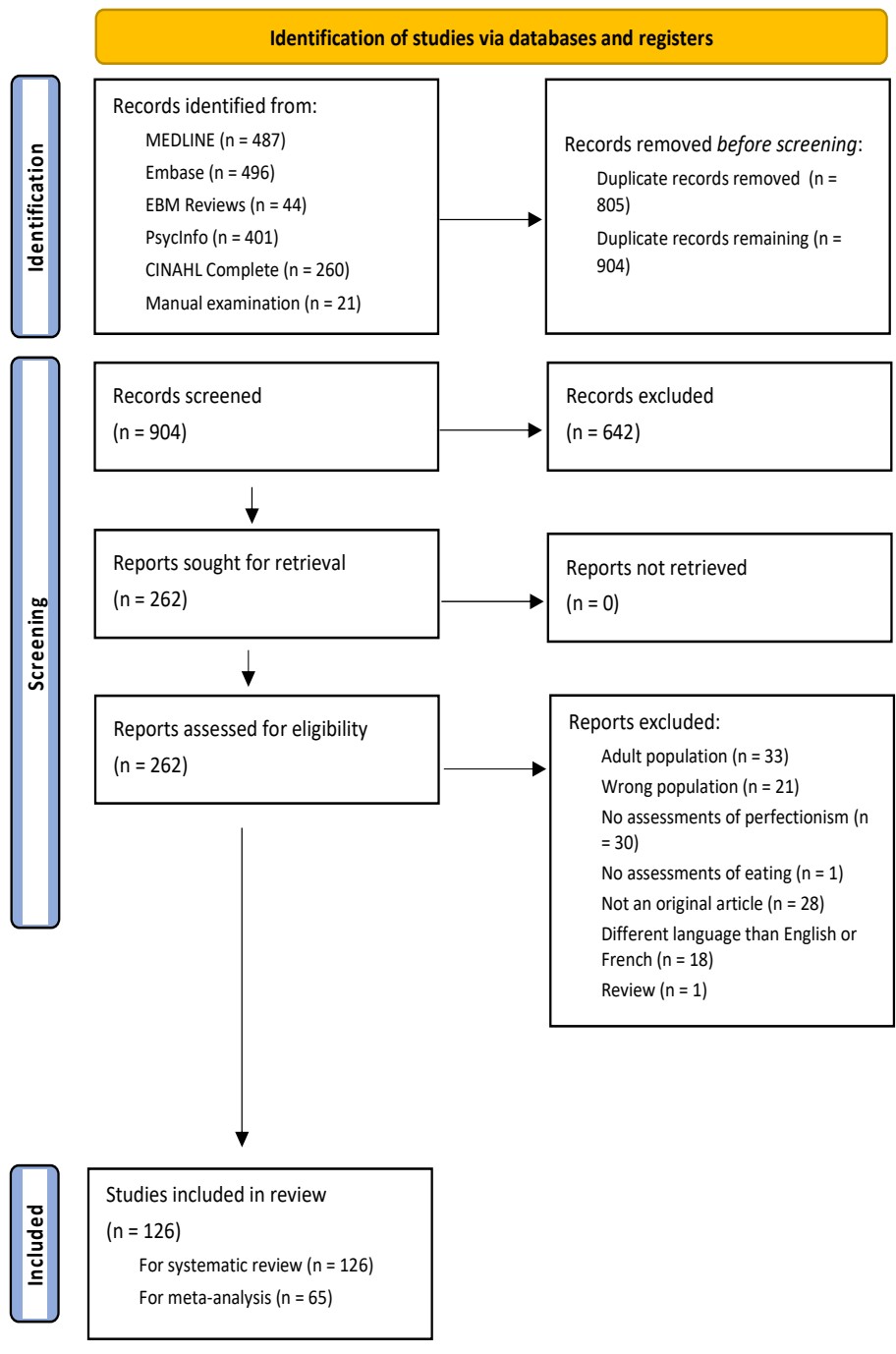

**Figure 1.** PRISMA 2020 Flow Diagram.

### 3.2. Magnitude of the Relationship between Perfectionism and Eating-Related Symptoms during Childhood and Adolescence

#### 3.2.1. Studies Assessing Eating Disorder Global Score

Meta-analyses using random-effects models (Figure 2) indicated small effect sizes between eating global scores and unidimensional perfectionism, perfectionistic strivings, and concerns ($r_{es}$ = 0.19, N = 14,197, k = 31, *p* < 0.001; $r_{es}$ = 0.21, N = 4961, k = 21, *p* < 0.001; $r_{es}$ = 0.13, N = 4105, k = 21, *p* < 0.001; respectively) with considerable heterogeneities (Figures 2 and 3, and Supplementary Table S6).

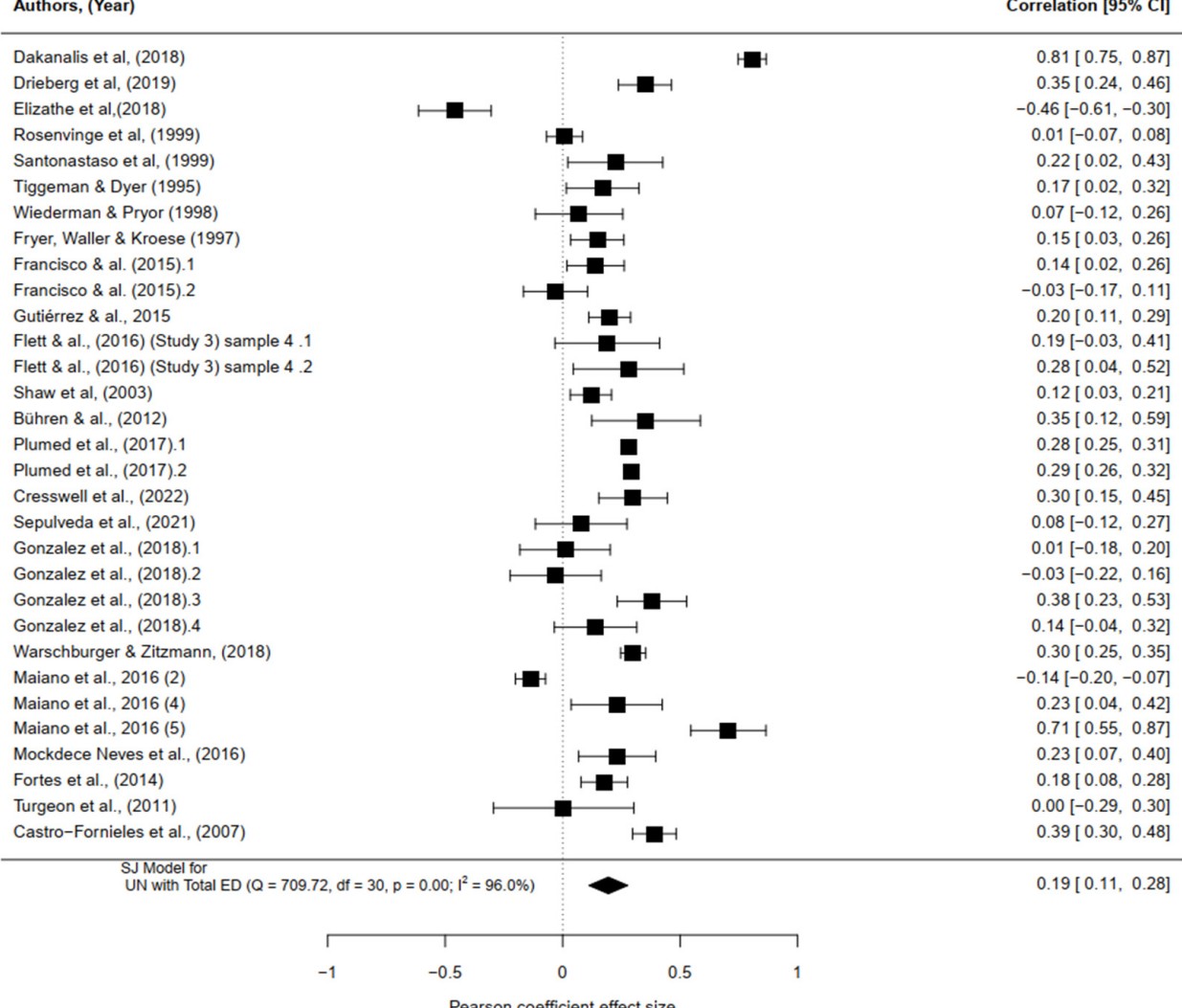

Unidimensional perfectionism/Eating global score

**Figure 2.** *Cont.*

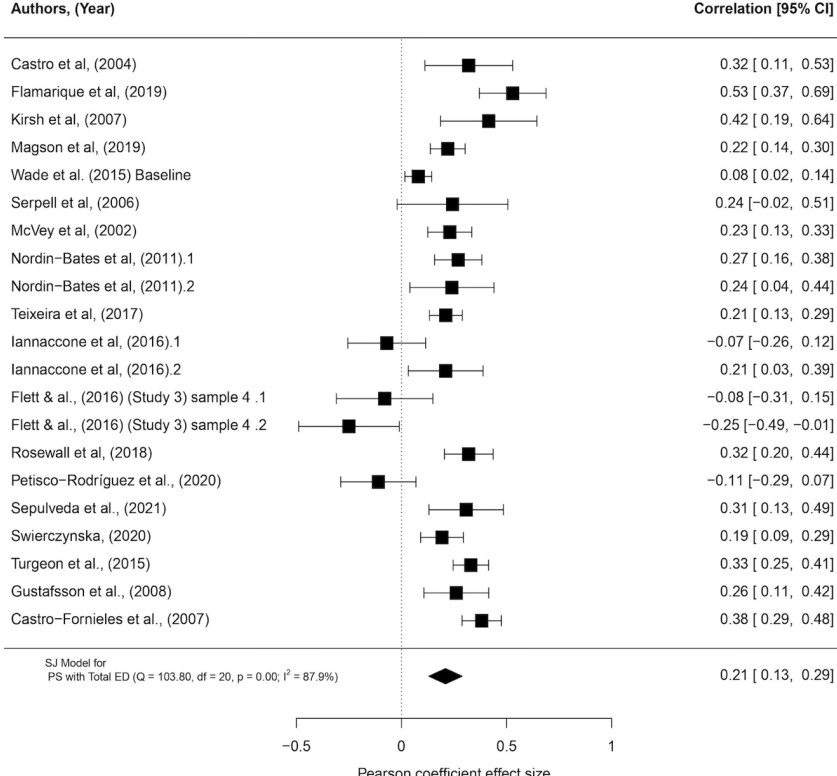

Perfectionistic strivings/Eating global score

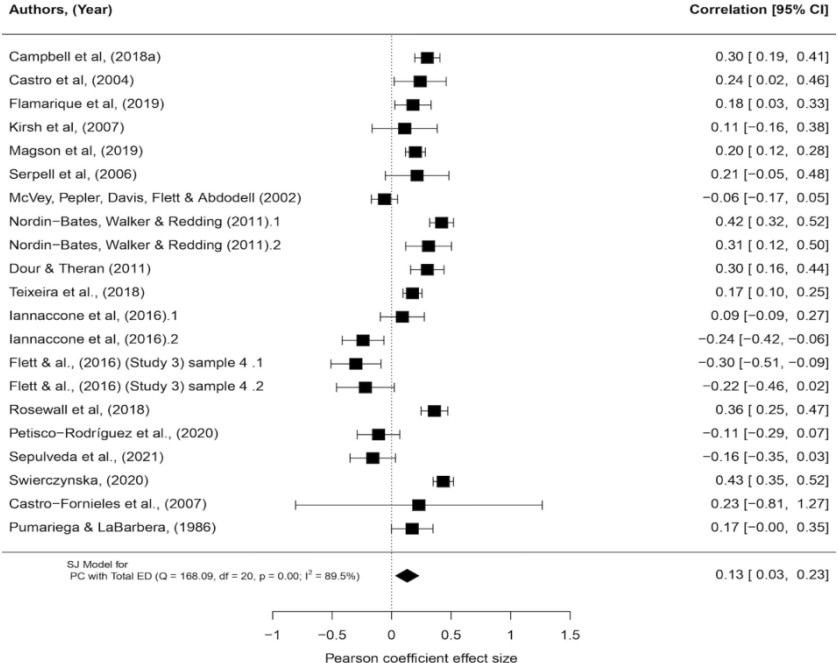

Perfectionistic concerns/Eating global score

**Figure 2.** Forest plot of the magnitude of the association between unidimensional perfectionism, perfectionistic strivings, perfectionistic concerns, and the eating global scores [12,20–23,27,49,51–84].

Note: Included samples are presented on the left of the forest plots, with 95% confidence intervals around Pearson's correlation effect sizes on the right. Squares represent the original studies' correlation effect size surrounded by the 95% confidence intervals. Diamonds represent the random-effect meta-analytic effect sizes. $I^2$ statistics quantify the heterogeneity in the random-effect meta-analyses.

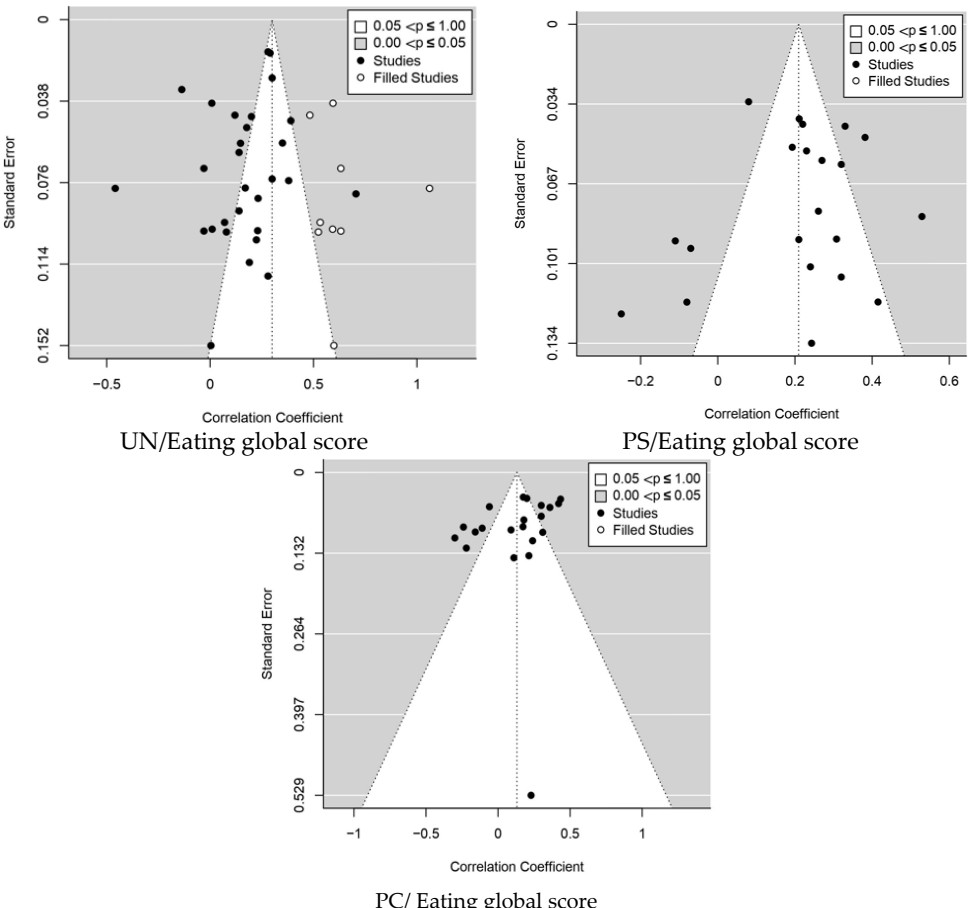

**Figure 3.** Funnel plots of the magnitude of the association between eating global scores, unidimensional perfectionism, perfectionistic strivings, and concerns.

Note: PS: Perfectionistic Striving; PC: Perfectionistic Concern; UN: Unidimensional Perfectionism. Funnel Plot with Trim and Fill for Global Eating Disorders total scores. Included studies are represented as black circles, whereas filled studies are represented as white circles in the plot to estimate the number of missing studies with extreme results. The publication bias was evaluated quantitatively with Egger's and Begg's tests.

3.2.2. Studies Assessing Eating-Related Symptoms

Unidimensional Perfectionism

Small effect sizes were observed between unidimensional perfectionism, binge eating, dietary restraint, body dissatisfaction, interoceptive awareness, internalization of thin ideal, drive for thinness, and weight and shape concerns ($r_{es}$ = 0.24, N = 3613, k = 10 studies, $p < 0.001$; $r_{es}$ = 0.27, N = 2061, k = 7 studies, $p < 0.001$; $r_{es}$ = 0.06, N = 1531, k = 3 studies, $p < 0.001$; $r_{es}$ = 0.25, N = 2298, k = 5 studies, $p < 0.001$; $r_{es}$ = 0.13, N = 3739, k = 9 studies, $p < 0.001$; $r_{es}$ = 0.22, N = 2874, k = 6 studies, $p < 0.001$; $r_{es}$ = 0.17, N = 1431, k = 4 studies, $p < 0.001$; respectively) with considerable heterogeneities (Supplementary Table S6, Figures S1 and S4).

Perfectionistic Strivings

Small effect sizes were observed between perfectionistic strivings, binge eating, dietary restraint, body dissatisfaction, eating concerns, drive for thinness, and weight and shape concerns ($r_{es}$ = 0.19, N = 1954, k = 7 studies, $p < 0.001$; $r_{es}$ = 0.19, N = 1549, k = 6 studies, $p < 0.001$; $r_{es}$ = 0.11, N = 1746, k = 6 studies, $p < 0.001$; $r_{es}$ = 0.18, N = 1510, k = 4 studies, $p < 0.001$; $r_{es}$ = 0.11, N = 772, k = 3 studies, $p < 0.001$; $r_{es}$ = 0.16, N = 3723, k = 7 studies, $p < 0.001$; respectively), from substantial to considerable heterogeneity (Supplementary Table S6, Figures S2 and S4).

Perfectionistic Concerns

Moderate effect sizes were found between perfectionistic concerns, binge eating, dietary restraint, drive for thinness, eating concerns, body dissatisfaction, and weight and shape concerns ($r_{es} = 0.37$, N = 2492, k = 6 studies, $p < 0.001$; $r_{es} = 0.32$, N = 1165, k = 3 studies, $p < 0.001$; $r_{es} = 0.31$, N = 772, k = 3 studies, $p < 0.001$; $r_{es} = 0.25$, N = 1510, k = 4 studies, $p < 0.001$; $r_{es} = 0.30$, N = 1687, k = 7 studies, $p < 0.001$; $r_{es} = 0.33$, N = 3723, k = 7 studies, $p < 0.001$; respectively), from moderate to considerable heterogeneities (Supplementary Table S6, Figures S3 and S4).

3.2.3. Subgroups Analysis

Associations in Community Samples

In community samples, unidimensional perfectionism was positively associated with a small effect size with the Eating global score ($r_{es} = 0.16$, N = 12,223, k = 20 studies, $p < 0.001$), binge eating ($r_{es} = 0.16$, N = 3107, k = 6 studies, $p < 0.01$), weight and shape concerns ($r_{es} = 0.23$, N = 1039, k = 2 studies, $p < 0.001$), body dissatisfaction ($r_{es} = 0.06$, N = 1531, k = 3 studies, $p < 0.001$), internalization of the thin ideal ($r_{es} = 0.19$, N = 3617, k = 8 studies, $p < 0.001$), interoceptive awareness ($r_{es} = 0.25$, N = 2298, k = 5 studies, $p < 0.001$), and drive for thinness ($r_{es} = 0.22$, N = 2874, k = 6 studies, $p < 0.001$), from substantial to considerable heterogeneity. A moderate association was found between unidimensional perfectionism and dietary restraint ($r_{es} = 0.35$, N = 1677, k = 4 studies, $p < 0.001$), with considerable heterogeneity (Supplementary Table S7, Figures S6 and S7).

In community samples, perfectionistic strivings were positively associated with a small effect size with the Eating global score ($r_{es} = 0.17$, N = 3248, k = 14 studies, $p < 0.001$), binge eating ($r_{es} = 0.17$, N = 1570, k = 4 studies, $p < 0.001$), dietary restraint ($r_{es} = 0.20$, N = 1116, k = 2 studies, $p < 0.001$), weight and shape concerns ($r_{es} = 0.15$, N = 3674, k = 6 studies, $p < 0.001$), body dissatisfaction ($r_{es} = 0.12$, N = 1746, k = 6 studies, $p < 0.001$), drive for thinness ($r_{es} = 0.11$, N = 772, k = 3 studies, $p < 0.001$), and eating concerns ($r_{es} = 0.20$, N = 1461, k = 3 studies, $p < 0.001$), from small to considerable heterogeneity (Supplementary Table S7, Figures S6 and S8).

In community samples, perfectionistic concerns were positively associated with a small effect size with the Eating global scores ($r_{es} = 0.13$, N = 3242, k = 14 studies, $p < 0.001$), with considerable heterogeneity. Perfectionistic concerns were moderately associated with binge eating ($r_{es} = 0.33$ N = 2492, k = 6 studies, $p < 0.001$), dietary restraint ($r_{es} = 0.33$ N = 1116, k = 2 studies, $p < 0.001$), weight and shape concerns ($r_{es} = 0.34$ N = 3674, k = 6 studies, $p < 0.001$), body dissatisfaction ($r_{es} = 0.30$ N = 1687, k = 7 studies, $p < 0.001$), drive for thinness ($r_{es} = 0.31$ N = 772, k = 3 studies, $p < 0.001$), and eating concerns ($r_{es} = 0.34$ N = 1461, k = 3 studies, $p < 0.001$), from substantial to considerable heterogeneity (Supplementary Table S7, Figures S6 and S9).

Associations in Clinical Samples

In clinical samples, unidimensional perfectionism was positively associated with a small effect size with the Eating global score ($r_{es} = 0.25$, N = 490, k = 3 studies, $p < 0.001$), dietary restraint ($r_{es} = 0.13$, N = 384, k = 3 studies, $p < 0.001$), weight and shape concerns ($r_{es} = 0.10$, N = 392, k = 2 studies, $p < 0.001$), from substantial to considerable heterogeneity. In addition, unidimensional perfectionism was moderately associated with binge eating ($r_{es} = 0.37$, N = 506, k = 4 studies, $p = 0.03$), with substantial heterogeneity (Supplementary Table S7, Figures S6 and S7).

In clinical samples, perfectionistic strivings were positively associated with a small effect size with the Eating global score ($r_{es} = 0.26$, N = 386, k = 4 studies, $p < 0.001$), binge eating ($r_{es} = 0.24$, N = 384, k = 3 studies, $p < 0.001$), dietary restraint ($r_{es} = 0.17$, N = 433, k = 4 studies, $p < 0.001$), from small to considerable heterogeneity (Supplementary Table S7, Figures S6 and S8).

Perfectionistic concerns were positively associated with a small effect size with the Eating global scores ($r_{es} = 0.17$, N = 231, k = 4 studies, $p < 0.001$), with small heterogeneity (Supplementary Table S7, Figures S6 and S9).

Associations in Samples with a Mean Age below 14

In samples with a mean age below 14 years old, unidimensional perfectionism was positively associated with a small effect size with the Eating global score ($r_{es}$ = 0.08, N = 5860, k = 14 studies, $p < 0.001$), binge eating ($r_{es}$ = 0.23, N = 2017, k = 5 studies, $p < 0.01$), body dissatisfaction ($r_{es}$ = 0.15, N = 1240, k = 2 studies, $p < 0.001$), internalization of the thin ideal ($r_{es}$ = 0.02, N = 1817, k = 5 studies, $p < 0.001$), interoceptive awareness ($r_{es}$ = 0.13, N = 968, k = 2 studies, $p < 0.001$), drive for thinness ($r_{es}$ = 0.24, N = 2283, k = 4 studies, $p < 0.001$), from small to considerable heterogeneity. Additionally, we observed a strong association between unidimensional perfectionism and dietary restraint ($r_{es}$ = 0.52, N = 486, k = 2 studies, $p < 0.001$), with considerable heterogeneity (Supplementary Table S7, Figures S6 and S7).

In both samples with a mean age below 14 years old, perfectionistic strivings were positively and associated with a small effect size with the Eating global score ($r_{es}$ = 0.24, N = 2231, k = 5 studies, $p < 0.001$), binge eating ($r_{es}$ = 0.15, N = 1132, k = 3 studies, $p < 0.001$), dietary restraint ($r_{es}$ = 0.22, N = 774, k = 2 studies, $p < 0.001$), and weight and shape concerns ($r_{es}$ = 0.17, N = 2017, k = 3 studies, $p < 0.001$), body dissatisfaction ($r_{es}$ = 0.04, N = 1014, k = 2 studies, $p < 0.001$), from small to considerable heterogeneities (Supplementary Table S7, Figures S6 and S8).

In samples with a mean age below 14 years old, perfectionistic concerns were positively associated with a small effect size with the Eating global score ($r_{es}$ = 0.14, N = 1036, k = 4 studies, $p < 0.001$). Perfectionistic concerns were moderately and positively associated with binge eating ($r_{es}$ = 0.38, N = 1580, k = 3 studies, $p < 0.001$), weight and shape concerns ($r_{es}$ = 0.41, N = 2037, k = 3 studies, $p < 0.001$), with considerable heterogeneity (Supplementary Table S7, Figures S6 and S9).

Associations in Samples with a Mean Age above 14

In samples with a mean age above 14 years old, unidimensional perfectionism was positively associated with a small effect size with the Eating global score ($r_{es}$ = 0.29, N = 2535, k = 15 studies, $p < 0.001$), binge eating ($r_{es}$ = 0.25, N = 2916, k = 5 studies, $p < 0.001$), dietary restraint ($r_{es}$ = 0.18, N = 1575, k = 5 studies, $p < 0.001$), weight and shape concerns ($r_{es}$ = 0.27, N = 1309, k = 3 studies, $p < 0.001$), and drive for thinness ($r_{es}$ = 0.18, N = 591, k = 2 studies, $p < 0.001$), with substantial considerable heterogeneity. We found moderate and positive effect sizes between unidimensional perfectionism, internalization of the thin ideal ($r_{es}$ = 0.36, N = 1039, k = 2 studies, $p < 0.001$), and the lack of interoceptive awareness ($r_{es}$ = 0.32, N = 1330, k = 3 studies, $p < 0.001$), from substantial to considerable heterogeneity (Supplementary Table S7, Figures S6 and S7).

In both samples with a mean age above 14 years old, perfectionistic strivings were positively associated with a small effect size with the Eating global score ($r_{es}$ = 0.17, N = 2306, k = 14 studies, $p < 0.001$), binge eating ($r_{es}$ = 0.23, N = 822, k = 4 studies, $p < 0.001$), dietary restraint ($r_{es}$ = 0.16, N = 775, k = 4 studies, $p < 0.001$), weight and shape concerns ($r_{es}$ = 0.16, N = 542, k = 2 studies, $p < 0.001$), body dissatisfaction ($r_{es}$ = 0.20, N = 690, k = 3 studies, $p < 0.001$), and eating concerns ($r_{es}$ = 0.14, N = 705, k = 2 studies, $p < 0.001$), from small to considerable heterogeneity (Supplementary Table S7, Figures S6 and S8).

In both samples with a mean age above 14 years old, perfectionistic concerns were positively associated with a small effect size with the Eating global score ($r_{es}$ = 0.10, N = 2618, k = 14 studies, $p < 0.001$), binge eating ($r_{es}$ = 0.28, N = 912, k = 3 studies, $p < 0.001$), dietary restraint ($r_{es}$ = 0.28, N = 509, k = 2 studies, $p < 0.001$), weight and shape concerns ($r_{es}$ = 0.28, N = 509, k = 2 studies, $p < 0.001$), body dissatisfaction ($r_{es}$ = 0.38, N = 787, k = 3 studies, $p < 0.001$), and eating concerns ($r_{es}$ = 0.14 N = 509, k = 2 studies, $p < 0.001$), from small to considerable heterogeneity (Supplementary Table S7, Figures S6 and S9).

### 3.3. Publication Bias

We observed a potential publication bias in the meta-analysis on the relation between unidimensional perfectionism and internalization of the thin ideal (Begg's test Kendall's tau = $-0.61$, $p = 0.0247$, Egger's test $p = 0.0013$), based on seven samples. We also observed a publication bias for the association between the Eating global score and both perfectionistic strivings and concerns in community samples (Begg's test Kendall's tau = $-0.1868$, $p = 0.3880$, Egger's test $p = 0.0212$; Begg's test Kendall's tau = $-0.3407$, $p = 0.1010$, Egger's test $p = 0.0034$) (Supplementary Figures S5 and S6).

## 4. Discussion

First, in the present study, we addressed notable gaps in the literature by synthesizing the findings of 126 studies and by testing concurrent associations between perfectionism and eating-related symptoms during childhood and adolescence. The first aim of the current systematic review was to extract and interpret data from 126 published studies on perfectionism in children and adolescents evaluating eating-related symptoms by self-report questionnaires and/or clinical evaluations. The originality of the current study also depends on the examination of the magnitude of the association between perfectionism and eating-related symptoms during childhood and adolescence in 65 studies. The systematic review included 86 studies using unidimensional measures of perfectionism and 87 of multidimensional perfectionism. We hypothesized that unidimensional perfectionism, perfectionistic concerns and perfectionistic strivings would be positively associated with all investigated eating-related symptoms. The results of our meta-analyses using (31 studies on 14,197 participants) confirmed our first hypothesis by indicating significant, small, and positive effect sizes between unidimensional perfectionism and the global score of eating measures. Additionally, unidimensional perfectionism was significantly, positively associated with a small effect size with the seven investigated eating measures. During childhood and adolescence, unidimensional perfectionism grows in the same direction as eating symptoms, especially binge eating, dietary restraint, a lack of interoceptive awareness, the internalization of thin ideal, drive for thinness, and weight and shape concerns. The current work empirically confirms the narrative findings of Vacca and her collaborators regarding the relevance of unidimensional perfectionism in eating pathology during adolescence [29].

The recent review of Vacca and her collaborators of 79 studies also indicated that 11 studies supported a relationship between EDs symptoms and perfectionistic concerns during adolescence and that only 5 studies reported these associations with perfectionistic strivings [29]. However, this narrative review based its conclusions on the number of associations found in the literature; there is no empirical evidence using a powerful statistical approach. Our results, based on the inclusion of the analysis of 126 articles, corroborate these previous findings but also present new findings. Here, the multidimensional approach to perfectionism was employed in 87 included studies. The relationships between Eds-related symptoms and perfectionistic strivings were found in 26 studies included in the meta-analysis, and in 30 studies included for perfectionistic concerns in the meta-analysis. Among studies that used the multidimensional assessment of perfectionism, the majority supported the association between ED symptoms, perfectionistic concerns and strivings [79,80,85]. The current work aimed and succeeded in determining the magnitude of these associations using a meta-analytical approach and confirmed our first hypothesis. A small effect size was found between perfectionistic strivings, the eating global score ($r_{es}$ = 0.21, N = 4961, k = 21, $p < 0.001$), and the other-dimensional measures of eating (i.e., binge eating, dietary restraint, eating concerns, weigh and shape concerns, body dissatisfaction, and drive for thinness). The pursuit of unrealistic objectives is associated with greater binging or/and restriction behaviors, but also more thoughts related to body image, thinness, weight and shape concerns, eating concerns, and body dissatisfaction. Regarding perfectionistic concerns, a small and positive association was also found with the Eating global score ($r_{es}$ = 0.13, N = 4105, k = 21, $p < 0.001$). Although this association with the Eating global score is higher in magnitude for PS rather than PC, all measures of eating symptoms are significantly, positively, and moderately associated with PC (i.e., binge eating, dietary restraint, weight and shape concerns, drive for thinness, and body dissatisfaction). During childhood and adolescence, the fear of making mistakes and being judged negatively by others was strongly associated with eating behaviors (binge eating, dietary restriction) and cognition related to body image like body dissatisfaction, weight, and shape concerns, eating concerns, and drive for thinness. The current finding is consistent with the body of research that suggests the strength of the relationship between PC and EDs is greater than between PS and EDs [29,86]. Perfectionistic concerns may offer one important avenue for advancing our understanding of how eating symptoms develop in youths.

Second, special attention was given to the investigation if all the discovered associations were sensitive to clinical status by stratifying extracted samples. We compared if these associations are stronger in clinical samples as established in the literature [25–34]. In both clinical and community samples, the Eating global score was significantly, positively, and associated with a small effect size with all types of investigated perfectionism (i.e., unidimensional, and perfectionistic concerns and strivings). Therefore, this overall association was higher in magnitude in clinical samples. Children and adolescents who are perfectionists are more prone to exhibit clinically relevant eating symptoms. Nevertheless, there are only a few studies in clinical samples evaluating the Eating global score. Further studies are warranted to enlarge the statistical power of this empirical evidence.

In community samples, unidimensional perfectionism was associated with a small effect size in relation to eating measures (i.e., binge eating, weight and shape concerns, body dissatisfaction, drive for thinness, internalization of the thin ideal, and the lack of interoceptive awareness). Interestingly, the association with binge eating was of higher magnitude in clinical samples. On the contrary, both associations with dietary restraint and weight and shape concerns were higher in community samples. During childhood and adolescence, binge eating appeared to be more likely in individuals struggling with perfectionism and an eating disorder. On the contrary, both dietary restrictions and weight and shape concerns are more likely to be used by perfectionist individuals within the community. Although associations are present between unidimensional perfectionism and other eating-related symptoms (i.e., body dissatisfaction, drive for thinness, internalization of the thin ideal, and the lack of interoceptive awareness), there is no study investigating these relationships in clinical samples. Further studies are warranted in childhood and adolescence to determine if these eating measures are more prone to be exhibited by perfectionist youth with an eating disorder.

In community samples, perfectionistic striving was associated with binge eating, dietary restraint, weight and shape concerns, body dissatisfaction, drive for thinness, and eating concerns. Interestingly, a similar association was also found between binge eating and dietary restraint also in clinical samples. In this context, perfectionistic strivings might comply with the theory of the need for control and the feeling of self-efficacy of Eds in the use of restriction and binge-purge behaviors when they are emotionally distressed [87]. The pursuit of unrealistic objectives is more likely to be found in the community and with an eating disorder. We could not perform any other analysis to estimate any effect size regarding perfectionism strivings due to the absence of enough studies in the clinical literature. We recommend that future studies investigate the association between perfectionistic striving and body dissatisfaction, weight and shape concerns, drive for thinness, and eating concerns in clinical samples.

Most importantly, in community samples, moderate effect sizes were found between perfectionistic concerns and all available eating measures (i.e., binge eating, dietary restraint, weight and shape concerns, body dissatisfaction, drive for thinness, and eating concerns). The fear of making mistakes and being judged negatively by others is very likely to be associated with eating symptoms in children or adolescents inside the community and at a higher magnitude than the pursuit of unrealistic objectives. In this context, high concerns over possible failures might maintain the feeling of not being sufficiently qualified and might lead, if not targeted, to an EDs. Thus, the current findings highlighted the effect of perfectionism on eating-related symptoms in individuals in both clinical and community samples during childhood and adolescence. As we hypothesized, this study demonstrated that the associations between perfectionism and eating symptoms in both clinical and community samples are different. It also underlined the major implication of perfectionism concerns in eating symptoms, even in community samples, before any clinical diagnosis. As we reported a few studies evaluating the association between perfectionistic concerns and EDs symptoms during adolescence in clinical samples, but also major findings in community samples, we recommend that future studies investigate this relationship.

Third, the current work also assessed the strength of these associations across childhood and adolescence to estimate if the associations between perfectionism and eating-related symptoms were stronger in a certain period during development. We expected associations to be different between studies that included children and early adolescents (i.e., under 14 years of age) and middle-to-late adolescents (i.e., over 14 years of age). In both samples with a mean age under or above 14 years old, the Eating global score was associated with small effect sizes with all three types of perfectionism. For unidimensional perfectionism, the Eating global score was slightly higher in magnitude in samples with a mean age above 14 years old.

In both samples, unidimensional perfectionism and perfectionistic strivings were associated with small effect sizes with all investigated eating measures. Interestingly, the association between unidimensional perfectionism and dietary restrictions was higher in magnitude in samples with a mean age under 14 years old. On the contrary, the associations between unidimensional perfectionism and internalization of the thin ideal and the lack of interoceptive awareness were higher in magnitude in samples with a mean age above 14 years old. Before the age of 14, perfectionistic youth appeared to be more likely to exhibit dietary restrictions, whereas, after 14, they are more likely to struggle with the internalization of the thin ideal and interoceptive awareness. Further studies are warranted to determine any difference in terms of age regarding unidimensional perfectionism, body dissatisfaction, weight, and shape concerns.

Regarding perfectionistic concerns, moderate effect sizes were found for binge eating, and weight and shape concerns in samples with a mean age under 14 years old, whereas they were found to be small in samples with a mean age above 14 years old. Binge eating and weight and shape concerns are more prone to be associated with the fear of making mistakes or being judged negatively by others before the age of 14. Additionally, the association with body dissatisfaction was found to be moderate in samples with a mean age above 14 years old. Body concerns are already associated with perfectionistic concerns in early adolescence. The abovementioned findings confirmed our third hypothesis related to the age groups.

The present study represents the most rigorous and comprehensive test of the relationship between dimensions of perfectionism and eating symptoms during childhood and adolescence. A key strength of the present study was the comprehensiveness of the systematic review and the meta-analytic methodology to help empirically support the associations between types of perfectionism and eating-related symptoms among adolescents. In addition, to identifying specific dimensions of perfectionism that perform a role in the development of eating symptoms, the findings revealed that youths who have strong evaluative concerns are more likely to develop eating-related symptoms during that critical period of childhood and adolescence. This study also demonstrated the stronger incidence of perfectionistic concerns in community samples and in samples with a mean age under 14 years old. While this meta-analysis offers insight into the current state of the literature, there are limitations associated with this study. First, the studies that were included were conducted among samples of mainly Western European females (USA, UK, Italy, etc.), limiting the generalizability of the findings. Moreover, in the meta-analyses, most of our samples are from the community (N = 27,099), whereas clinical samples are underrepresented (N = 2169). This made our findings less applicable to clinical populations.

Second, many of the included studies were cross-sectional or consisted of uncontrolled pre-post trials with short-term follow-up periods. Therefore, conclusions about the temporal relation between perfectionism and eating disorder symptoms remain undetermined. Additionally, perfectionism cannot be clearly established as a significant risk factor for the development of eating disorders. Furthermore, no longitudinal meta-analysis was conducted since the primary studies did not collect data at measurement times in a systematic way. Further longitudinal studies are required to evaluate temporal precedence and the directionality between perfectionism and eating-related symptoms in adolescents, notably by performing multilevel modeling. In addition, more studies evaluating the moderating role of, rather than stratifying participants according to, sex and gender on the perfectionism eating-related symptoms relationship are required to evaluate potential sex differences.

Reviewing the literature and extracting data led to some limitations. Similarly, in the meta-analysis on the same topic in adulthood led by Limburg and his collaborators in 2017 [19], the sample sizes were relatively small for some measures due to the few available studies on the topic, affecting the power of some analyses. Some analyses also presented publication bias (See Section 3.3). There was also considerable heterogeneity observed within samples (Tables S6 and S7 in supplementary materials), which may have been influenced by the clinical and/or methodological differences witnessed across the studies. Indeed, the current work demonstrates the variety of instruments used to assess perfectionism. This study reported five different instruments to measure unidimensional perfectionism and five other instruments to estimate dimensional perfectionism (See Table S3 in supplementary materials). We recommend further studies to systematically report standardized summary statistics from questionnaires, and we encourage studies to use well-recognized questionnaires to reduce heterogeneity between studies (see Section 3.1. for a list of the most reported questionnaires). Additionally, this work pointed out that there is a lack of studies, including EDs clinical samples on perfectionistic concerns and studies including children.

Thus, some of the results need to be interpreted with caution due to the small sample sizes. Future research is required to examine longitudinal effects over longer follow-up periods to elucidate the role of perfectionism in the development of eating symptoms, including the contributing roles of the different types of perfectionism. Future research should also clearly distinguish between the findings supporting perfectionism as a risk factor for eating disorders (longitudinal) and findings demonstrating an association between perfectionism which could become more present as a result of the development of the illness itself, and eating disorder symptoms (cross-sectional). Recent findings in the literature demonstrated reciprocal relations between perfectionistic concerns, depressive or anxious symptoms, obsessive-compulsive disorders, and suicidal ideations in the adolescent and adulthood literature [19,88–91]. Another area for future research consideration is to examine the added role of potential covariates such as internalizing symptoms (i.e., depression and anxiety), obsessions, suicidal ideas, or other personality types to determine if they exert a confounding influence (i.e., a mediating or moderating role). Thus, additional research is warranted to examine more complex models by considering mediation and moderation effects. As a recent meta-analysis demonstrated that perfectionistic concerns represent a complex interlocking condition, which influences each other in a vicious circle [89], we encourage clinicians to evaluate and treat depressive symptoms as they mutually influence each other.

In the present study, the variables of perfectionism and eating disorder symptoms were assessed via self-report measurement, which can be influenced by social desirability. The number of dimensions of the multidimensional approach of perfectionism makes it difficult to study. We found associations between unidimensional perfectionism, even in the subgroup analyses. Our findings indicate that measuring unidimensional perfectionism with disordered eating could be sufficient in studies with limited room for assessment. Nevertheless, since the multidimensional approach detects specific facets of perfectionism involved in the expression of EDs [19,29], this approach pushes things further and allows the capture of diverging effects of perfectionism dimensions on ED symptoms across time and samples.

In the current study, only two dimensions of perfectionism were considered (perfectionistic strivings and perfectionistic concerns). However, perfectionistic concerns consist of six facets (socially prescribed perfectionism, evaluative concerns, concerns over mistakes, doubts about actions, parental expectations, and parental criticism), and four compose perfectionistic strivings (self-oriented perfectionism, other-oriented perfectionism, personal standards, and organization). Future studies are warranted to study how subtypes of perfectionism can perform a role in the development of ED and the onset of perfectionistic traits. There is also a need to examine the reciprocal association between PS and PC during childhood and adolescence and the interaction between perfectionism and eating-related symptoms in predicting ED outcomes.

The overall prevalence of EDs among adolescent samples ranges from 4.5% to 8.5% among females, and from 1.2% to 2.2% among males. Females also have a 10% lifetime risk of clinical Eds [3,92–94]. Some studies conducted on adults found that perfectionism predicted ED symptoms for women but not for men [95]. Nevertheless, there is no evidence in the literature regarding gender differences in the relationships between perfectionistic dimensions and eating symptoms among adolescents. Most of the studies included girls in their samples, and some used mixed gender samples (87% in clinical samples and 68% in community samples). The current meta-analysis was not performed in a sex-specific manner since most of the studies excluded adolescent males. Additionally, studies which included both adolescent males and females did not provide sex-specific results to avoid reducing sample sizes. Moreover, the directionality in the perfectionism-eating symptoms could not be assessed because we focused on existing correlation results. No meta-regression was performed because information regarding confounding factors was not available across primary studies. More research is needed to shed light on sex differences in the associations between perfectionism and eating symptoms during childhood and adolescence.

The current work has important clinical implications regarding the targeted prevention of perfectionism in children and adolescents in clinical or community settings. Some recent meta-analyses have demonstrated an increase in perfectionism levels across time and that perfectionism is associated with adverse mental health outcomes, and that perfectionism is a transdiagnostic process across anxiety, depression, obsessive-compulsive disorders, and eating disorders [50,79,80,85]. Another recent meta-analysis on 15 randomized controlled trials demonstrated that CBT for perfectionism was effective in reducing perfectionism (with large effect size), symptoms of depression (medium effect size), anxiety (small effect size), and eating disorders (medium effects) [79].

The current findings emphasize the importance of targeting perfectionism in individuals with and without any diagnosis of EDs. More specifically, associations were established between binge eating, dietary restraint, weight and shape concerns, and unidimensional perfectionism in both samples. A higher magnitude was found in clinical samples for binge eating, whereas the opposite was found for dietary restraint, weight and shape concerns. We recommend clinicians track the relationship between dietary restraint, weight and shape concerns, and perfectionism in individuals without any diagnosis of EDs and regularly evaluate if the association is evolving into the direction of clinically relevant EDs. Additionally, we also recommend clinicians systematically evaluate the presence of perfectionism in individuals with EDs using binge eating strategies.

The current study also emphasizes the importance of targeting perfectionism early in the development phase as perfectionism and its dimensions (strivings and concerns). As the associations between binge eating and weight and shape concerns and perfectionistic concerns were higher in magnitude in samples with a mean age under fourteen, we recommend the early prevention of eating-related problematic behaviors and thought patterns to target specific concerns about being judged negatively by others.

## 5. Conclusions

The findings of the present study confirm that children and adolescents who strive to be perfect and report having concerns around perfectionism are more prone to exhibit eating-related symptoms during childhood and adolescence. Perfectionistic concerns appear to be related to eating-related symptoms with a higher magnitude. We recommend targeting perfectionistic concerns with early prevention strategies before the age of fourteen and inside the community. Targeting perfectionism concerns early in development may also prevent the emergence of EDs and other psychopathologies.

**Supplementary Materials:** The following supporting information can be downloaded at: https://www.mdpi.com/article/10.3390/adolescents3020022/s1. Table S1: PRISMA Checklist, Table S2: Search strategies for the five databases, Table S3: Studies included in the systematic review, Table S4: Number of studies included in meta-analyses after data extraction, Table S5: Instruments reported in the literature and measuring perfectionism in childhood and adolescence, Table S6: Summary of meta-analyses using random-effects models on the relationship between eating related symptoms total scores and perfectionism types, Table S7: Summary of meta-analyses using random-effects models on the relationship between eating related symptoms total scores and perfectionism (subdivided by their clinical status and their mean age. Figure S1: Forest plot of the magnitude of the association between unidimensional perfectionism and eating related symptoms, Figure S2: Forest plot of the magnitude of the association between perfectionistic strivings and eating related symptoms, Figure S3: Forest plot of the magnitude of the association between perfectionistic concerns and eating related symptoms, Figure S4: Funnel plots for the association between eating related symptoms, unidimensional perfectionism, perfectionistic strivings and concerns, Figure S5: Forest plot of the magnitude of the association between unidimensional perfectionism, perfectionistic strivings, perfectionistic concerns and eating global scores (subdivided by their clinical status and their mean age), Figure S6: Forest plot of the magnitude of the association between unidimensional perfectionism and eating related symptoms (subdivided by their clinical status and their mean age), Figure S7: Forest plot of the magnitude of the association between perfectionistic strivings and eating related symptoms (subdivided by their clinical status and their mean age), Figure S8: Forest plot of the magnitude of the association between perfectionistic concerns, eating related symptoms (subdivided by their clinical status and their mean age). Refs [96–228] are cited in the Supplementary file.

**Author Contributions:** A.L. designed the study. A.L. and S.C. executed PRISMA protocols and performed literature searches. A.L. and F.M.Y. made the quality assessment of the included studies. A.L. and P.P.P. collected and extracted data from primary articles and created tables. X.N. created figures and performed statistical analyses. A.L. wrote the first draft of the manuscript. G.M., N.C. and P.C. made comments. All authors collaborated on data interpretation and manuscript draft revisions. All authors have read and agreed to the published version of the manuscript.

**Funding:** This work was supported by CIHR operating grant FRN114887, ERA-NET Neuron from CIRH/FRQS FRN 278649/NDD161472, the Eating Disorders Ontario University Health Network, and the Canada Research Chair to support Dr. Conrod's salary.

**Institutional Review Board Statement:** Not applicable.

**Informed Consent Statement:** Not applicable.

**Data Availability Statement:** The data presented in this study are available in Figures 1–3, and Supplementary Materials.

**Acknowledgments:** Conrod was supported by a senior investigator award from the Tier 1 Canada Research Chair. Livet was supported by a grant awarded to PC from the Canadian Institutes of Health Research. All authors thank Nguyen Julian-Khôi-Huu, a resident in psychiatry, at the University of Montreal, and Poulin Camille, a sophomore student at Georgetown University, for their help in the initial phase of the selection of articles during their internship at the Conrod Venture Lab (Summer 2019 and 2020). All authors thank Julien Ouellet, a candidate in Neurosciences at the University of Montreal, for his help in checking English grammar and spelling. We thank all the reviewers for their time and efforts that have been spent, and for their valuable suggestions, which contributed to the final version of the article.

**Conflicts of Interest:** The authors declare no conflict of interest.

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
