# Peer review of "Perfectionism in Children and Adolescents with Eating-Related Symptoms: A Systematic Review and a Meta-Analysis of Effect Estimates"

_adolescents, doi:10.3390/adolescents3020022_

Round 1

Reviewer 1 Report

Comments and Suggestions for Authors

I'd like to thank the editor for the opportunity to review this manuscript. The authors have performed a very impressive work, including a lot of papers and performing both a systematic review and a meta-analysis of the data. The paper applied robust methodology, and they reported their work according to the PRISMA guidelines - as suggested by the literature. I do not have specific concerns, but I think this paper might be very helpful in the field. I have some comments for the authors, but they are more suggestions for implementing the paper.

- I cannot find the quality evaluation of the included paper. This might be a relevant aspect that might help the analysis because it can help to exclude papers with low quality.

- I think the discussion might be improved by splitting the clinical implication into a separate sub-chapter from the general discussion. This might help clinicians to focus on these results, while researchers might be more interested in the methodology or questionnaires applied (but this is really a suggestion). 

- Have you performed a quality check of the papers' inclusion/exclusion and data extractions? 

- I think the authors might include more explicit suggestions for the future studies, especially regarding questionnaires or approaches.

Author Response

Dear reviewer,

            Many thanks for taking time to review our submitted article for consideration for inclusion in your special issue of Adolescents entitled “Perfectionism in children and adolescents with eating related symptoms: a systematic review and a meta-analysis of effect estimates”. Our sincere thanks to the reviewers for sharing their comments and thoughts which we found very useful. As suggested, we have revised the manuscript using MS Word Track Changes, and have addressed each issue raised, point-by-point below.

Thank you again for the valuable feedback. We hope that you and the reviewers will find the revised paper substantially improved and now acceptable for publication in Adolescents.

Reviewer 2 Report

Comments and Suggestions for Authors

The current review is of eating disorders and perfectionism in adolescents. I appreciate the authors' focus on an important topic, but several aspects of the report dampened my enthusiasm for the paper. In particular, the introduction does not adequately introduce the study or its findings, the results are difficult to follow due to the reporting of numerous associations without clear organization of findings, and the discussion mainly reiterates findings. I have specified several areas for improvement below.

Introduction

·      Very long and perhaps overly detailed. Discussion of measures used and prevalences of specific EDs does not seem necessary. I would tighten to include brief (1-2 sentence) description of prevalence and harms of ED in adolescents, mention of importance in identifying risk factors (1 sentence), and then introduce perfectionism by with definition and overlap with EDs.

·      Perfectionism paragraph is also very general and overly detailed. Please tighten to focus on definition of perfectionism and theory behind it, along with associations with ED and theory of why it is associated with ED.

·      Why was 14 decided as the mean age to cut off the samples? I understand it was the mean age of the samples but any theoretical reason why a 14 year old and 15 year old would differ?

·      More theory and evidence on the differences between perfectionistic strivings and perfectionistic concerns is needed, given the emphasis on the differences with each with eating disorders symptoms reported in the results.

·      More discussion on community and clinical samples needed, given the emphasis on comparing them in the results. Why would we expect them to differ? What evidence or theory is there to support this?

·      More justification is needed for what this study contributes to the literature beyond the 2020 European Eating Disorders Review systematic review. The authors mention considerations of magnitude are needed, but does this review add anything else?

·      What were the authors hypotheses?

Methods

·      Did the authors ensure that studies not focused on elite athletes excluded elite athletes? If not, how do they know samples were not biased?

·      Why were only English and French articles included?

·      The literature review was completed on December 9? The authors completed all of the work for this article in roughly 2 months? Is this correct? That is a rather quick timeline

Results

·      Both 3.2.3.3 and 3.2.3.4 are labeled “Comparison in clinical samples” – this is confusing. I am further confused by what it all even means – are the authors comparing the age groups or clinical and community samples to each other?

·      What is meant by publication bias?

·      Overall, the results are long, cumbersome to read through, and confusingly presented. I suggest a table or something to simplify for the reader.

·      The language “weak” effect size is not scientific. Please revise to use terms generally used in assessing effect sizes – small, medium, large.

Discussion

·      The Discussion largely just reiterates the findings of the study, as opposed to placing the findings in the context of previous empirical findings and theory and providing recommendations for future research.

·      The authors report in the discussion (for what seems like the first time) that associations are greater in community samples – why might that be? Seems counterintuitive as clinical samples are typically reporting more severe psychopathology

·      The authors report they tested cross-lagged reciprocal associations, but most studies were cross-sectional so this language is confusing. How can the authors test reciprocal relations for assessments that occurred at the same time?

·      How were inclusion criteria different from the Vacca et al review? Why were more studies included in this review?

Author Response

Dear reviewer,

            Many thanks for taking time to review our submitted article for consideration for inclusion in your special issue of Adolescents entitled “Perfectionism in children and adolescents with eating related symptoms: a systematic review and a meta-analysis of effect estimates”. Our sincere thanks to you for sharing your comments and thoughts which we found very useful. As suggested, we have revised the manuscript using MS Word Track Changes, and have addressed each issue raised, point-by-point below.

Thank you again for the valuable feedback. We hope that you will find the revised paper substantially improved and now acceptable for publication in Adolescents.

Round 2

Reviewer 2 Report

Comments and Suggestions for Authors

The authors were not responsive to my concerns about the length and level of detail in the introduction - it is unnecessarily detailed.

Despite agreeing in their response that "weak" is not appropriate for describing effect size, the authors continue to use "weak" throughout the results and discussion, rather than "small". I recommend the authors do a thorough read-through or justify why they are using subjective, rather than objective, adjectives to describe their effect size findings. 

The discussion is extremely long and it is very easy to get lost in the details. I recommend the authors cut it down substantially and focus on the major points rather than reiterating all the details of their findings.

Author Response

Manuscript ID: adolescents-2228299

Perfectionism in children and adolescents with eating related symptoms: a systematic review and a meta-analysis of effect estimates.

Dear reviewer,

            Many thanks for taking time to review our submitted article for consideration for inclusion in your special issue of Adolescents entitled “Perfectionism in children and adolescents with eating related symptoms: a systematic review and a meta-analysis of effect estimates”. Our sincere thanks to you for sharing your second review which we found very useful. As suggested, we have revised the manuscript using MS Word Track Changes, and have addressed each issue raised, point-by-point below. Thank you again for your valuable feedback.

Comment 1 “The authors were not responsive to my concerns about the length and level of detail in the introduction - it is unnecessarily detailed”.

Response: We thank the reviewer for pointing out that our introduction was still too detailed. We deleted all the sentences related to the measures used to assess perfectionism or EDs and all sentences in relation to the prevalences of specific EDs.

Comment 2 “Despite agreeing in their response that "weak" is not appropriate for describing effect size, the authors continue to use "weak" throughout the results and discussion, rather than "small". I recommend the authors do a thorough read-through or justify why they are using subjective, rather than objective, adjectives to describe their effect size findings.

Response: We would like to thank the reviewer for his/her rigorous examination of our manuscript. We also agree that weak was not an appropriate word regards to effect size. We revised the use of the word ‘weak’ for qualifying effect sizes and replaced it by the suggested appropriate term “small” when required throughout the manuscript.

Comment 3 “The discussion is extremely long, and it is very easy to get lost in the details. I recommend the authors cut it down substantially and focus on the major points rather than reiterating all the details of their findings.

Response: In the updated version of the discussion, we decided to focus on the major points of our results rather than reiterating all the details of our findings. We divided the discussion section in three parts which are related to our three main hypotheses. We placed our findings in the context of previous empirical findings and theory and provided recommendations for future research.